# The Role of Extracellular Vesicles as Modulators of the Tumor Microenvironment, Metastasis and Drug Resistance in Colorectal Cancer

**DOI:** 10.3390/cancers11060746

**Published:** 2019-05-29

**Authors:** Kodappully S. Siveen, Afsheen Raza, Eiman I. Ahmed, Abdul Q. Khan, Kirti S. Prabhu, Shilpa Kuttikrishnan, Jericha M. Mateo, Hatem Zayed, Kakil Rasul, Fouad Azizi, Said Dermime, Martin Steinhoff, Shahab Uddin

**Affiliations:** 1Academic Health System, Translational Research Institute, Hamad Medical Corporation, Doha P.O. Box 3050, Qatar; SSivaraman@hamad.qa (K.S.S.); emoibrahim04@gmail.com (E.I.A.); AKhan42@hamad.qa (A.Q.K.); KPrabhu@hamad.qa (K.S.P.); SKuttikrishnan@hamad.qa (S.K.); Jmateo1@hamad.qa (J.M.M.); FAzizi@hamad.qa (F.A.); MSteinhoff@hamad.qa (M.S.); 2National Center for Cancer Care and Research, Hamad Medical Corporation, Doha P.O. Box 3050, Qatar; Araza@hamad.qa (A.R.); Krasul@hamad.qa (K.R.); SDermime@hamad.qa (S.D.); 3College of Health Sciences, Department of Biomedical Sciences, Qatar University, Doha P.O. Box 2713, Qatar; hatem.zayed@qu.edu.qa; 4Department of Dermatology Venereology, Hamad Medical Corporation, Doha, P.O. Box 3050, Qatar; 5Weill Cornell-Medicine, Doha P.O. Box 24811, Qatar; 6Weill Cornell University, New York, NY 10065, USA

**Keywords:** colorectal cancer, extracellular vesicles, tumor microenvironment, metastasis, drug resistance, miRNA, biomarker

## Abstract

Colorectal cancer (CRC) is one of the most common cancers worldwide, with high morbidity and mortality rates. A number of factors including modulation of the tumor microenvironment, high metastatic capability, and resistance to treatment have been associated with CRC disease progression. Recent studies have documented that tumor-derived extracellular vesicles (EVs) play a significant role in intercellular communication in CRC via transfer of cargo lipids, proteins, DNA and RNAs to the recipient tumor cells. This transfer influences a number of immune-related pathways leading to activation/differentiation/expression of immune cells and modulation of the tumor microenvironment that plays a significant role in CRC progression, metastasis, and drug resistance. Furthermore, tumor-derived EVs are secreted in large amounts in biological fluids of CRC patients and as such the expression analysis of EV cargoes have been associated with prognosis or response to therapy and may be a source of therapeutic targets. This review aims to provide a comprehensive insight into the role of EVs in the modulation of the tumor microenvironment and its effects on CRC progression, metastasis, and drug resistance. On the other hand, the potential role of CRC derived EVs as a source of biomarkers of response and therapeutic targets will be discussed in detail to understand the dynamic role of EVs in CRC diagnosis, treatment, and management.

## 1. Introduction

Colorectal cancer (CRC) is the third most common cancer globally, with high morbidity and mortality rates. Worldwide, approximately 9% of all cancer incidence and 8% of all cancer deaths have been attributed to CRC, with approximately 1.2 million new CRC cases diagnosed per year [1,2]. The incidence of CRC has been reported to be highest in developed countries, including Australia, New Zealand, Canada, the United States and parts of Europe [3].

The natural course of CRC development involves alterations due to well-defined histopathological changes (adenoma-carcinoma sequence) that initiate tumor development leading to tumor progression. According to the American Joint Committee on Cancer (AJCC), CRC has been grouped into four distinct stages of transition. In stage I, CRC cells are actively proliferating, forming small, benign polyps. A small fraction of these polyps accumulates various somatic mutations over time leading to the transformation of benign polyps into malignant types, thus transitioning into stage II. In stage III, the tumors cells gradually increase in number, accumulating further mutations with the invasion of the surrounding muscle wall of the colon. Finally, in stage IV, these cancer cells accomplish the capability to metastasize through the blood and the lymphatic system to other organs. However, modulation of the tumor microenvironment/immunological pathways in the tumor landscape has also been associated to play a significant role in CRC development [4].

The main options for CRC treatment include surgery and chemotherapy, depending mainly on the CRC stage and the location of the polyps [5]. Substantial advancements have been made in the treatment and management of CRC, mainly through the administration of adjuvant chemotherapy agents like fluorouracil and oxaliplatin [6]. Unfortunately, until now, the success rate of chemotherapy in CRC is only 30% [7,8]. Though the treatment modalities in CRC have advanced considerably in recent years, the main challenge that clinicians continue to endure is drug resistance that serves as an impediment for CRC management [9]. On the other hand, survival rates in CRC depend largely on the stage at which disease is diagnosed, with 5-year survival rates observed to range from approximately 90% for stage I to 10% for stage IV disease [2]. Thus, the earlier the diagnosis, the higher the chance of survival, indicating that diagnostic/prognostic biomarkers may play in prominent role in efficient screening and management of CRC [4]. Recently, studies have demonstrated that tumor cells secrete membrane-bound vesicles, called extracellular vesicles (EVs), that serve as efficient mediators of intercellular communication [10] by releasing large amounts of nucleic acids, cytokines/chemokines, angiogenic factors, extracellular matrix remodeling factors as well as tumor microenvironment modulating factors during carcinogenesis [11,12,13]. Studies have also documented that cargoes transferred through EVs can alter the functions and phenotypes of recipient cells [14,15]. In CRC, this is important as these alterations can affect the regulation of multiple cellular processes [16], lead to modulation of the tumor microenvironment and enhance tumor cell proliferation and transformation [14,15]. On the other hand, in addition to exerting cellular changes, EVs are secreted abundantly in body fluids such as plasma, serum, saliva, urine, cerebrospinal fluid (CSF), breast milk, bile, broncho-alveoler lavage fluid, and malignant ascites. It is well perceived that due to these intrinsic properties, CRC-derived EVs are being actively explored as a potential source of new prognostic/diagnostic biomarkers [17].

The main aim of this review is to focus on the role of EVs in CRC with respect to immune modulation, drug resistance, and metastasis, as well as to review the literature regarding EVs as a potential source of therapeutic targets in CRC.

## 2. EVs and Colorectal Cancer

According to the MISEV2018 guidelines, extracellular vesicles (EVs), is a collective term covering subtypes of cell-released, membranous nanometre-sized structures that are classified into various subpopulations based on size, biogenesis and cell origin or function [18]. Based on sizes, EV subpopulations include nanovesicles, microvesicles, virus-like particles, exosome-like vesicles, and microparticles, while on the basis of biogenesis, exosomes, membrane particles, outer membrane vesicles, and shedding membrane vesicles are considered important subpopulations. Furthermore, platelet-dust, oncosomes, matrix-vesicles, ectosomes, dexosomes, texosomes, epididymosomes, cardiosomes, prostasomes, rhinosomes, apoptotic bodies, and tolerosomes are another set of subpopulations that are described on the basis of cell origin or function. However, with this exceptional diversity in EV subpopulations, it has been reported that even further subpopulations within these subpopulations can be defined, based on vesicle size, density, RNA, protein, and DNA cargo, as well as morphology [19]. For the sake of clarity in this review, we will consider the following EV subtypes for the purpose of discussion: (a) Small EVs, 30–150 nm in size, secreted via a multivesicular-body endocytic process and designated as exosomes; and (b) microvesicles, 100–1000 nm in size, formed via outward budding of the plasma membrane [20,21].

Comparison of small EVs/exosomes with microvesicles has shown that small EVs/exosomes are more homogenous with respect to size and molecular composition. However, due to the differences in the origin, the mechanism of formation and release, exosomes and microparticles often have differences in the levels of cargo proteins [19]. Due to the variances in isolation methods and characterization protocols used by researchers, it is very difficult to pinpoint a single cargo content that can clearly distinguish exosomes and microvesicles [22]. Since both the exosomes and microvesicles are membrane covered particles released by the cells to facilitate intracellular communication and have a key role in several pathological conditions, they are combined under the family of extracellular vesicles.

Structurally, the phospholipid bilayer of EVs is composed mainly of lipids, including ceramide, cholesterol, sphingomyelin, phosphoglycerides, glycosphingolipids, phosphatidyl serine (PS), phosphatidylethanolamine, mannose, N-linked glycans, polylactosamine and sialic acid [13]. Studies have reported that there are a number of proteins that originate from the cytosol or endosomal compartments along with a few proteins from plasma membrane in EVs that are considered as potential markers for EV sub-populations. These include tetraspanins (CD9, CD63, CD81 and CD82), 14-3-3 proteins, major histocompatibility complex (MHC) molecules and cytosolic proteins such as heat shock proteins, Tsg101 and the Endosomal Sorting Complex Required for Transport (ESCRT-3) binding protein Alix, Rab GTPases, SNARE, flotillins, Annexin, Clathrin, and platelet-derived growth factor receptors, etc. [23].

Studies have reported that EVs are produced in high quantities by rapidly growing cells including B-lymphocytes, dendritic cells, cytotoxic T cells, platelets, mast cells, neurons, oligo-dendrocytes, Schwann cells and intestinal epithelial cells [24]. Therefore, an increase in the amount and alterations in the expression of EV cargoes are widely reported to indicate pathological conditions, including cancers [16]. This is evidenced by studies that have shown that tumor cell-derived EVs promote cancer progression and invasion by various mechanisms. A study on the highly malignant melanoma cell line, B16-10, and poorly metastatic F1 melanoma cell lines showed that the EVs secreted by malignant melanoma cells were able to transfer their metastatic ability efficiently to poorly metastatic tumor cells, indicating that EVs can induce phenotypic changes to enhance the metastatic capabilities of tumor cells [25]. On the other hand, EVs have also been associated with incurring an anti-tumor potential, e.g., NK cells derived EVs containing both perforin and CD95L, have been reported to express anti-melanoma activity both in vitro and in vivo [26]. Similarly, dendritic cell-derived EVs (Dex), loaded with MHC class I and II-restricted cancer antigens, have been reported to serve as effective maintenance therapy in non-small cell lung cancer (NSCLC) via boosting of natural killer (NK) cell arm of antitumor immunity [27].

In CRC, the role of EVs in tumor development and proliferation has been studied extensively with interesting results. A study on HT29-19, a permanently differentiated clone of human colorectal adenocarcinoma cell line HT29, has reported that EVs secreted by HT29-19, from apical and basolateral sides show distinct morphological features [28]. The EVs secreted from the apical sides were found to be homogenous, with approximately three-fold higher protein content per million cells than the EVs secreted from the basolateral side. In comparison, the EVs secreted by the basolateral side were found to be often aggregated with heterogeneous size and shape. Irrespective of secretion from the apical and basolateral sides, the secreted EVs were found to contain MHC class I molecules, CD26/dipeptidyl peptidase IV (DPPIV), CD63 and CD26 [28]. This finding is significant as it provides an insight that the secreted EVs may carry accessory molecules involved in antigen presentation and thus can modulate the immune cells independent of direct cellular contact with effector cells [28]. Similarly, a study on EVs isolated from culture supernatants of CRC cell line SW403 and peripheral blood of CRC patients were both found to contain Fas ligand and TRAIL on their membrane with CD63, HLA class I molecule and carcinoembryonic antigen (CEA) as its cargo proteins. On the other hand, EVs derived from peripheral blood of healthy controls did not express any of these proteins, indicating that CRC derived EVs are independently capable of inducing apoptosis in activated T-cells through Fas ligand and TRAIL [29].

EVs produced by CRC cell lines have also been reported to induce proliferation, migratory capacity, and invasiveness. For example, EVs secreted by CRC cell line, SW480, have been reported to express CEA, CD81 and TSG101, which has been documented to induce manifestations of tumor-like phenotypes, morphological changes, spheroid formation and localization of plasma membrane vacuolar H^+^-ATPase [30]. Likewise, human colon adenocarcinoma cell line, HCA-7, has also been reported to release EVs containing EGFR ligands-heparin-binding EGF-like growth factor (HB-EGF) and amphiregulin, that can lead to enhanced invasiveness in tumor cells [31]. Similar studies on the metastatic potential of EVs have been done with interesting results, e.g., study on human colorectal cancer cell lines, HCT-15, SW480, and WiDr, has reported that EVs secreted by these cell lines contain mRNAs (CD81), microRNAs (miR-21, miR-192, and miR-221), and natural antisense RNAs (LRRC24, MDM2 and CDKN1A genes) that are delivered into human hepatocellular carcinoma cell line HepG2 and human lung cancer cell line A549. The results from this study are relevant, as they reveal the capability of EVs to shuttle between cells and play a vital role in gene regulation and expression inducing a metastasis potential in recipient cells [32]. Furthermore, a study on CRC specific cell lines, SW480, has documented that SW480 derived EVs are efficiently transferred to HepG2 cells via dynamin and clathrin-dependent endocytosis leading to activation of mitogen-activated protein kinase pathway thus enhancing cell migration in the target cells [33].

Published literature on CRC EVs has documented that these EVs can induce bidirectional interactions among tumor and tumor-associated cells via modulation of the tumor microenvironment. Details on modulation of tumor microenvironment in CRC and the role of EVs are discussed in detail below.

## 3. Components of the Tumor Microenvironment in CRC

CRC is a highly heterogeneous disease that exhibits differences in its biological, clinical and physiologic functions. The major players in CRC development and progression include somatic changes induced by genetic and epigenetic mutations that allow cancerous cells to attain features required for survival and unrestrained proliferation [34]. In addition to these genetic alterations, a critical biological feature of CRC, that serves as a contributor to sustained growth and invasion is the ‘tumor microenvironment’ [35]. The tumor microenvironment (TME) by definition is an interactive surrounding cellular environment around the tumor comprising of blood vessels, immune cells, inflammatory cells, lymphocytes, signaling molecules, fibroblasts, bone marrow-derived and the extracellular matrix (ECM) [36]. The main function of TME is to establish a cellular communication pathway with diverse capacities to induce beneficial consequences for tumorigenesis such as inducing peripheral immune tolerance and releasing a series of extracellular/angiogenesis signaling molecules leading to extensive modulation of the TME. This leads to the generation of highly aberrant tissue functions. The modulated TME then converts into a pathological entity that continually evolves to aid cancer progression and invasion [36]. The major components of the tumor microenvironment include cancer-associated fibroblasts, immune cells and immune-suppressive cells that play a significant role in tumor survival and metastasis (Table 1).

The normal colon comprises of a stromal population that provides a physical architecture for uniform functioning of the colonic tissue. The main constituent of the colonic stroma includes fibroblasts that not only form the structural framework of tissues through their secretion of extracellular matrix constituents (ECM) but also aids in synthesis, deposition, and homeostasis of the basement membrane components thus maintaining tissue function [59]. Keeping the significance of colonic tissue integrity as a critical parameter for normal functioning of the colon, tumor cells aim to modulate and transform/differentiate fibroblasts into dysregulated cell types, known as cancer-associated fibroblasts (CAF) [60]. Once differentiated, CAF’s, release large number of growth factors, hormones and cytokines such as osteopontin (OPN), hepatocyte growth factors (HGF), epidermal growth factors (EGF), IGF 1 and 2, FGF 2 and 7, PGE-2, vascular endothelial growth factor (VEGF), stromal derived factor-1 (SDF-1), macrophage migration inhibitory factor (MIF), vitronectin, interleukins, microRNA (miRNA) that modulate immune responses to directly and indirectly support tumorigenesis. Furthermore, CAF’s also contribute to the initiation of motile and proteolytic phenotype in tumor cells by inducing epithelial-mesenchymal transition (EMT) that potentiates invasiveness and metastasis [60]. In CRC, CAFs are the most abundant stromal cells involved in modulation of the TME into a highly vascularized environment thus promoting tumor progression and leading to poor clinical outcomes [61].

In addition to CAFs, the CRC landscape is infiltrated by a variety of innate and adaptive immune cells, including lymphocytes (T cells, B cells, natural killer cells), monocytes, macrophages, dendritic cells, granulocytes (neutrophils, basophils, eosinophils, mast cells), T regulatory cells (Tregs) and myeloid-derived suppressor cells (MDSC) [62]. Of these immune cells, macrophages are considered to play an important role in immune defense by regulating T-and B-lymphocyte activation and proliferation. In CRC, release of chemokines, cytokines and growth/angiogenic factors such as monocyte chemotactic protein-1 (CCL2), CCL3, CCL4, CCL5 (RANTES), CCL22 (macrophage-derived chemokine), CXC chemokines (CXCL8), Colony-stimulating factor-1 (CSF1) etc. polarizes these macrophages to the tumor tissue converting them into highly specialized cells known as tumor-associated macrophages (TAMs) [63]. These TAMs, in turn, facilitate modulation of tumor microenvironment by initiating an immunosuppressive entity through the release of cytokines/chemokines and immunosuppressive factors such as IL-10, TGF-β, immune checkpoint modulators (Programmed death-1/programmed death ligand 1), Tregs, myeloid-derived suppressor cells (MDSCs) and type 2 helper (Th2) T cells. These released factors directly suppress activated T cells leading to enhanced tumor growth and survival [41].

In the tumor microenvironment, the most widely known immune-suppressive cells include myeloid-derived suppressor cells (MDSC) and the T regulatory cells (Tregs). The MDSC is functionally defined as immunosuppressive, immature myeloid cells that significantly contribute to tumor progression via angiogenesis and metastasis [64]. Activation of MDSC during tumorigenesis occurs via the release of factors either by activated T cells or tumor stromal cells including COX2, prostaglandins, IL-6, GM-CSF, VEGF, IFN-γ, IL-13, TGF-β. These factors stimulate myelopoiesis leading to upregulation of STAT3 target genes (B-cell lymphoma XL (BCL-XL), cyclin D1, MYC, survivin, S100 calcium binding protein A8 (S100A8) and S100A9) and signaling pathways (STAT6, STAT1, and NF-κB) allowing infiltration of MDSC to tumor sites [65]. Once infiltrated, MDSC exerts their immunosuppressive action by modulating/inhibiting mechanisms of immune-surveillance including antigen presentation by dendritic cells (DCs), T cell activation, M1 macrophage polarization and NK cell cytotoxicity [66]. In CRC, metastasis and tumor growth has been associated with elevated levels of circulating and tumor localized MDSCs that release cytokine/chemokine/ protein within the tumor microenvironment (CCL2, CCL15, S100A8, and S100A9) and activate signaling pathways such as MAPK and NF-κB for sustained growth [67,68].

Another set of immune-suppressive cells in the tumor microenvironment includes Tregs which have diverse effects on tumorigenesis. Tregs mainly exert immunosuppression by secreting immune-suppressive cytokines such as IL-10, TGF-β, and IL-35 and affecting cytotoxic T cell function by inhibiting cytolytic granule release. Furthermore, Tregs suppress the activation, proliferation and effector functions of CD4+ and CD8+ cells, NK cells, B cells, maturation of dendritic cells (DC) as well as inhibition of co-stimulatory molecules In addition to exerting immunosuppressive activity, infiltration of Tregs can initiate angiogenesis and increased vascularization leading to poor outcomes in CRC patients [43].

In addition to these immune suppressive cells, there are a variety of other immune cell populations found in the tumor microenvironment such as dendritic cells, monocytes, neutrophils, mast cells, NK cells, CD4, CD8 cells, B cells that play diverse roles in promoting tumorigenesis in CRC via degranulation, release of pro-angiogenic factors, growth stimulatory factors (VEGF, FGF2, TNF-α), angiopoitin-1 and tryptase, granzyme B, perforin and upregulation of MHC class 1 and co-stimulatory ligands leading to tumor progression and proliferation [61].

## 4. Role of EVs in Immune System Regulation and Modulation in CRC

EVs have been reported to regulate the immune system by activation of antigen-specific CD4+ or CD8+ T cell responses through antigen presentation via direct and indirect pathways (Figure 1). In the direct pathway, either the MHC class II peptide complexes, co-stimulatory (Tim1/4 for B cells) and adhesion molecules (ICAM-1) on the EV surface interact with corresponding T cell receptor to activate CD4+ or CD8+ cells or the EVs directly activate macrophages, neutrophils, natural killer cells, and APCs subsequently activating CD4+ or CD8+ T cells. In indirect pathway, either the EVs are internalized by antigen presenting cells (APCs), or they bind to the APCs via integrins and intercellular adhesion molecules leading to antigen presentation and activation of CD4+ or CD8+ T cells [42,69]. Due to the intrinsic ability of EVs to regulate the T cells, cancer-associated EVs can facilitate immune modulation and suppression within the tumor microenvironment to promote tumor progression. In CRC, immune suppression, immune escape and drug resistance within the microenvironment is mediated by CRC EVs utilizing various interaction such as (a) remodeling of tumor-stromal interactions, (b) transfer of genetic materials including proteins, mRNA and non-coding RNAs (miRNA, long coding RNA, etc.) and (c) release of MDSC and immune suppressive molecules [70].

Remodeling of tumor-stromal interactions in CRC occurs via transfer of stromal EVs in colonic mesenchymal stem cells (cMSC). This leads to major aberrations in cMSC including the formation of spheroids, atypical morphology and increased vacuolar H^+^-ATPase redistribution that induces an acidic microenvironment. All these remodeling associated changes exert an immune suppressive (due to acidic environment) and immune-privileged (due to spheroids) conditioned environment for the CRC cells allowing them to proliferate, invade and metastasize [30]. Furthermore, spheroid formation initiates NOTCH3/Wnt signaling pathways that also confer drug resistance associated properties to the CRC cells [45].

Another way of tumor-stromal interactions re-modeling in CRC is via EV-mediated transfer of non-coding RNAs (Table 2). Transfer of non-coding RNAs plays a critical role in inducing aberrant behaviors in tumors and modulation of tumor microenvironment functions. The main non-coding RNAs involved in immune modulation in CRC includes miRNAs (miR210, miR193a, miR19a, etc.), long coding RNA (MAGEA3, CRNDE-h) and circular RNA (circ-KLDHC10, circRTN4, etc.). The main mechanism of action is through the interaction of non-coding RNAs with their mRNA transcripts leading to deregulation of mRNA and converting them into oncogenes or tumor suppressors that promote tumor progression and metastasis. Furthermore, these deregulated mRNAs also influence the tumor microenvironment by increasing the proliferation of fibroblasts, epithelial and endothelial target cells within the microenvironment leading to immune suppression and induction of drug resistance [47,48].

The CRC EVs mediate immunosuppressive circuit and release of MDSCs by interacting with monocytes through membrane fusion and altering their differentiation towards transforming growth factor (TGF-β) secreting myeloid suppressor cells. These transformed myeloid cells exert inhibitory effects on T lymphocyte proliferation and effector functions by expressing CD14 on their surface with lack of co-stimulatory molecule up-regulation and reduced expression of HLA class II. This facilitates transformed myeloid cells to spontaneously secrete TGF-β—a key cell for immune suppression [123]. Furthermore, CRC EVs also promote immunosuppressive activity by autocrine stimulation of the IL-6/STAT3 pathway leading to immune suppression via secretion of MDSC population thus facilitating tumor growth. In addition to this, the CRC EVs mediate hyper activation of Wnt 4/β-catenin signaling due to hypoxic microenvironment, leading to enhanced proliferation and migration of endothelial cells. This accelerates tumor growth and promotes angiogenesis [61].

It is evident that the CRC EVs are involved in modulation of the tumor microenvironment and secretion of immune suppressive cells to facilitate proliferation, angiogenesis, invasion, and metastasis in colorectal cancer.

## 5. Role of EVs in the Induction of Metastasis in CRC

EVs, through their content specific molecules, have been identified as critical players in tumor metastasis, transformation, and invasion (Table 1). Studies have reported that EVs induce invasive and metastatic abilities in tumor cells via interactions with the cytoskeleton, extracellular matrix, transfer of non-coding RNAs and immune regulation [124,125].

Studies on cytoskeleton remodeling have reported that cancer cells are capable of forming actin rich protrusions known as invadopodia to initiate invasiveness. Furthermore, EVs have been reported to be associated with these invadopodia to facilitate ECM degradation and metastasis. The main mechanism of these invadopodia is to dock to the ECM, then recruit EVs to their plasma membrane allowing them to secrete matrix metalloproteinases that initiate degradation of the ECM and facilitate invasiveness [124]. On the other hand, it has been reported that mature invadopodia have the capability to facilitate further invasiveness by inducing invadopodia formation in non-invasive cells thus allowing enhanced secretion of proteinases via EVs and increasing invasiveness and metastatic potential of cancer cells [124]. In CRC, invadopodia formation with upregulation and secretion of several matrix metalloproteinases (MMP) have been associated with invasiveness and metastasis indicating critical role of cytoskeleton remodeling and associated EVs in promoting metastasis [126].

EVs also have the capability to interact with ECM directly to transfer cargoes that lead to ECM degradation. E.g., exosomes expressing high levels of CD96c and CD44 are capable of binding to the hyaluronic acid of the ECM leading to enhanced degradation and invasiveness [125]. Similarly, a study on CRC cell lines has reported that fibroblasts activated by late stage cancer-exosomes are capable of upregulating pro-invasive regulators of membrane protrusion (PDLIM1, MYO1B) and matrix-remodeling proteins (MMP11, EMMPRIN, ADAM10) leading to increased invasiveness via ECM [127]. Furthermore, a study on CRC patients has documented upregulation of MMP9, cathepsin B (CTSB), and A disintegrin and metalloproteinase with thrombospondin motifs 13 (ADAMTS13) in serum purified exosomes of CRC patients. As these MMPs are critical for ECM degradation as well enhancement of immune and signaling pathways such as TGF-β/Smad pathway, their upregulation indicates that the CRC exosomes play a vital role in promotion of adhesion as well as triggering of signaling pathways/inflammatory responses in cancer cells for metastasis and increased invasiveness [128].

In addition to these, an interesting feature of EVs is their ability to establish a pre-metastatic niche—a phenomenon whereby the primary tumor can promote its metastasis by recruiting immune cells to distant organs to establish a supportive metastatic environment [129]. In CRC, the liver has been reported to be the main target of metastatic organotrophism for the establishment of the pre-metastatic niche. It is reported that the establishment of the pre-metastatic niche is a critical step in the initiation of metastasis and is mediated by many factors. Firstly, the expression of integrins (cell receptor adhesion proteins) such as ITGα6β4/ITGα6β1 by CRC EVs serves as favorable determinants of metastatic organotropism due to their adhesion with specific resident cells and fibroblasts/epithelial cells conferring invasive properties to the CRC cells [49,50]. Secondly, a transcriptome analysis study on CRC mediated EVs and their effect on liver organotropism have shown that immune modulation within the tumor microenvironment can play an important role in proliferation and metastasis. Major players described in immune modulation and subsequent extracellular matrix remodeling includes macrophage infiltration, activation of Src phosphorylation signaling pathways (S100A families), the release of pro-inflammatory cytokines and chemokines (IL-6, TNF-α, CCL8, CXCL1, CXCL13, TGF-β) [53]. Thirdly, immune modulation in the tumor microenvironment to facilitate metastasis by CRC EVs is also reported to be a result of exposure of cMSC to EV components. This leads to the spheroid formation which allows functional changes in the microenvironment leading to higher proliferation, migration, and invasion [30]. In addition to these, another mechanism associated with migration and invasion as well as conditioning a permissive pre-metastatic niche for colonization in CRC cells is associated with the transfer of non-coding RNAs, in particular, miR17-92a, miR-210, miR-19a, miR-193a, etc. These non-coding RNAs serve to regulate macrophages skewing them towards a pro-inflammatory phenotype thus facilitating colonization and initiating invasive properties in these cells [30]. Furthermore, proteomics analysis has shown that upregulation of a number of CRC mediated proteins such as SERPINA1, SERPINF2 and MMP9 play an important role in the extracellular matrix and cytoskeleton remodeling leading to vascular leakiness and tumor-promoting migration and invasiveness [51]. Therefore, it is concluded that a large number of CRC EV-associated non-coding RNA, proteins and inflammatory molecules serve as key players in tumor proliferation, the establishment of pre-metastatic niche, invasion, and metastasis.

## 6. Role of EVs in Drug Resistance in CRC

Drug resistance in CRC has also been attributed to the EVs through their content specific cargoes. The main mechanism of drug resistance via CRC EVs is through sequestering of cytotoxic drugs in their vesicles and subsequent expulsion [54]. Exosomes do this by sequestration of cytotoxic drugs in the intracellular vesicles and subsequent expulsion, to negate drug effect within the cells [130,131]. A study in melanoma, adenocarcinoma and lymphoma cells reported that drugs such as cisplatin, 5-flurouracil and vinblastin were confiscated in the endosomal compartment of these cancer cells [132]. Similarly, exosomes secreted from cisplatin resistant ovarian cancer cells showed 2.6-fold higher drug concentration indicating the role of exosomes in extrusion of cytotoxic drugs. On the other hand, rate of exosome secretion and its composition was found to be altered in cancer cells exposed to radiation indicating their role in radiotherapy resistance as well [133]. Furthermore, EVs have been documented to carry drug efflux pumps such as P-glycoprotein 1 also known as multidrug resistance protein 1 (MDR1) or ATP-binding cassette sub-family B member 1 (ABCB1), ABCG2 or ABCA3 that facilitate the transfer of drug resistance to recipient cancer cells [134]. Several studies on cancer models of prostate, ovarian, leukemia and osteosarcoma have reported the role of EVs carrying these drug efflux pumps in the transfer of multidrug resistance to sensitive cells indicating this as one of the mechanisms utilized by EVs in conferring drug resistance in recipient cells [135,136,137,138]. It has been documented that EVs are capable of transferring these MDR proteins, either as a fully functional protein or as mRNA encoding for the protein, by direct interaction with the recipient cancer cell [138].

Another common mechanism of drug resistance documented for EVs is via transfer of miRNAs that cause alteration of EMT mediated signaling pathways such as TGF-β/SMAD and targeting mRNA’s that subsequently lead to up regulation of drug-efflux pumps [139]. In a study on colorectal cell lines, it was observed that CRC exosomes are capable of activating Wnt/β-catenin pathway by promoting the stabilization and nuclear translocation of β-catenin leading to 5-FU and oxaliplatin resistance [56]. On the other hand, EVs can also induce drug resistance via targeting of apoptosis regulators such as Bcl-2/BAX signaling pathways that transfer drug resistance in recipient cells via promotion of anti-apoptotic pathways [140]. Furthermore, reports on augmentation of drug resistance pathways via modulation of apoptotic pathways have been described in stromal and breast cancer cells. In this study it was observed that transfer of non-coding RNA via EVs to cancer cells lead to activation of the pattern recognition receptor RIG-I that in turn activated STAT1-dependent anti-viral signaling. Simultaneously, the stromal cells induced activation of NOTCH3 on breast cancer cells. This paracrine anti-viral and juxtacrine NOTCH3 pathways converge, leading to facilitation of STAT1 transcriptional responses to NOTCH3 thus expanding therapy resistant breast cancer cells [58]. Similarly, in CRC cell lines, EVs have been associated with cetuximab resistance via down regulation of PTEN leading to increased phosphorylation of AKT levels [141]. Many studies have documented another mechanism of drug resistance by EVs. It has been reported that as a result of treatment such as hypoxia or radiation, levels of TNF-α, TGF-β, PDGF, CXCL12, MMP, and HIF are elevated within the tumor associated EVs. These EVs carrying cancer stem cell features, when transferred to recipient cancer cells induce modulation of various signaling molecules such as Hedgehog, Wnt, β-catenin etc., thus enhancing resistance to cancer therapies [142]. On the other hand, chemoresistance in CRC is also attributed to the ability of miRNAs to promote stemness in cancer stem cells (CSC) leading to atypical morphology and spheroid formation [57].

A recent study on EVs derived from both serum and tissue of CRC patients was found to be enriched in miR-196b-5p as compared to the healthy control subjects. Comparison in survival analysis indicated that CRC patients having high levels of miR-196b-5p had poor survival. The main reason for this was associated with targeting of immune regulators, SOCS1 and SOCS3, of STAT3 signaling pathway leading to activation of STAT3 signaling thus promoting stemness and chemoresistance against 5-fluorouracil [57]. Further studies on this aspect have reported that certain tumor markers such as CK19, TAG72, and CA125 are expressed by EVs in chemotherapy-resistant and highly metastatic CRC cells indicating EVs can be utilized as chemoresistance markers in CRC [143].

## 7. EVs as Potential Source of Biomarkers in CRC

In CRC, it is well established now with many studies showing promising results that EVs can serve as potential source of biomarkers of response, recurrence or drug-resistance (Figure 2). Elevated levels of EV cargoes, especially miRNAs, lncRNA, circ. RNAs, in biological samples of CRC patients, have been linked to the high expression of cancer antigens and shorter survival [144]. The most valuable EV derived candidates in CRC diagnostics include miRNAs like miR-21, miR-29a, miR-92a, miR-100, miR-200, miR-223, miR-1229 [51,117,145,146,147]. Studies have reported that high expression of miRNAs, such as miR-125a-3p and miR21 can be useful for diagnosing a patient at an early stage or before undergoing any treatments [148]. Similarly, EVs like miR-19a and miR-4772-3p have been reported in high quantities in human serum samples and can serve as a potential prognostic marker for disease recurrence in CRC [149].

Furthermore, analysis of serum samples of CRC patients have shown the presence of collapsing response mediator protein, along with EVs, and these have been directly correlated to poor survival rates [151,152,153]. On the other hand, proteomic analyses of EVs from ascites of CRC patients have reported the expression of endosomal markers such as ephrin-B1 and cadherin-17 [154].

A recent study conducted to determine whether glypican 1 (GPC1) positive EVs in plasma samples could be used as a potential biomarker for CRC has shown interesting results. Higher levels of GPC1 in CRC patients were observed before surgery as compared to normal healthy controls. In addition to GPC1, non-coding RNAs, miR-96-5p and miR-149, were also identified as potential candidates for diagnosis of CRC [155].

Similar to miRNAs, long coding RNAs (lncRNAs) secreted by EVs are also being investigated as a biomarker in CRC [52]. Studies on bio-informatics analysis have identified Bladder Cancer Associated Transcript 1 (BLACAT1) and lncRNAs (LOC344887, LINC00675, DPP10-AS1, HAGLR) to be downregulated in CRC patients as compared to healthy volunteers [156]. Similarly, another study has identified mRNAs (KRTAP5-4, MAGEA3) and lncRNA (BCAR4) as potential candidates for the detection of CRC [121]. Similarly, circular RNAs were found to be downregulated in HCT116 and KRAS mutant cells compared to cells bearing wild type KRAS [157] suggesting its use as a biomarker in the screening of CRC patients bearing KRAS mutation. On the other hand, a new cirRNA (cir-KLDHC10) was significantly upregulated in CRC-derived EVs in comparison to EVs of healthy participants indicating its potential as a biomarker [158].

The main topic of interest now is to utilize these EVs for drug delivery and therapeutics. Studies have shown that engineering of EVs can facilitate delivery of drug and therapeutic nucleic acids precisely. E.g., HCT 116 cells engineered to express miR-379 within EVs, when transferred to recipient cancer, decreased proliferation and migration rate of CRC cells [159]. Similarly, a study conducted in a colon adenocarcinoma mouse model reported that EVs delivering doxorubicin were able to reduce tumor size more efficiently in comparison to free or liposome-delivered doxorubicin [160]. This indicates that EVs not only play a significant role in the CRC tumorigenesis process but also serve as a potential source of biomarkers or therapeutic targets, that can be modelled into a useful tool for the clinical management of CRC patients.

## 8. Challenges for Using EVs as Potential Source of Biomarkers in CRC

The role of EVs in the modulation of the tumor microenvironment, metastasis and drug resistance with promising evidence on their utility as potential source of biomarkers in CRC has been explored in detail in this review. However, with various studies exploring the translational potential of EVs, a number of limitations and challenges need to be addressed in order to prevent undesirable effects with its application. Firstly, an efficient way of EVs classification has not been achieved as yet. As molecular profiles of EVs have been observed to be different within the same pathogenesis, it is suggested that until a robust method of EVs classification based on protein profiling, nucleic acid profiling, size or affinity are described, its use as potential source of biomarkers should be utilized with caution [51]. Secondly, detection technologies for distinguishing EVs have not been properly standardized [17]. The main methods utilized for EV isolation include ultracentrifugation, microfiltration, gel filtration, precipitation with polyethylene glycol, protamine, and sodium acetate [22]. Recently, microfluidic technologies for isolation of EV population based on highly specific interactions with the molecules exposed on the EV surface have also bene defined. However, due to technical hurdles in each of these isolation methods such as complexity, loss of sample, failure to separate large vesicles with similar sedimentation rates, contamination with viral particles, deformation of vesicles, small quantity of exosomal proteins, co-isolation of large protein aggregates/lipoproteins, poor reproducibility, contamination with non-EV proteins, aggregation in multi-vesicles and difficulties with detachment of molecules/analysis of intact vesicles have been reported, it is critical that standardization with robust internal and external quality control assessments need to be in place in order to validate their isolation so that they can be utilized as potential source of biomarkers [22]. Thirdly, large randomized control trials/translational studies need to be performed so that validation of their efficient therapeutic potential can be determined [161]. This is critical as EVs carry signature proteins from the host and therefore their application in non-host patients may have immunogenic implications [162]. Therefore, with standardized detection/isolation techniques and profiling methods along with robust clinical trials in place, the utility of EVs in CRC and other cancer could be highly valuable for therapeutic implications and as potential source of biomarkers.

## 9. Conclusions

Evidence documented to date indicates that EVs are capable of inducing various modulations in the tumor microenvironment leading to metastasis, tumor cell proliferation, and drug resistance. Utilizing this knowledge, it is possible to understand the dynamics of EVs in immune modulation with CRC therapeutic utility. This strategy would include engineering EVs for vaccine development and induction of immuno-modulatory mechanisms to promote anti-tumor responses, such as depletion of Tregs and MDSCs, increasing proliferation of NK cells, monitoring of cytokine/chemokines and utilizing EV cargoes as nanoparticles for drug delivery. Furthermore, EVs can also serve as novel non-invasive markers that can be used as diagnostic tools for screening and management of CRC. However, it is recommended that the data on CRC derived EVs should be extrapolated to extensive large, randomized clinical trials to further validate the published data. Furthermore, translational studies need to be initiated to ascertain the therapeutic potential and utility of EVs. Future perspectives should be EV-based immunotherapy, EVs as delivery vectors, or EVs as therapeutic targets in CRC that will help in cancer management.

## Figures and Tables

**Figure 1 cancers-11-00746-f001:**
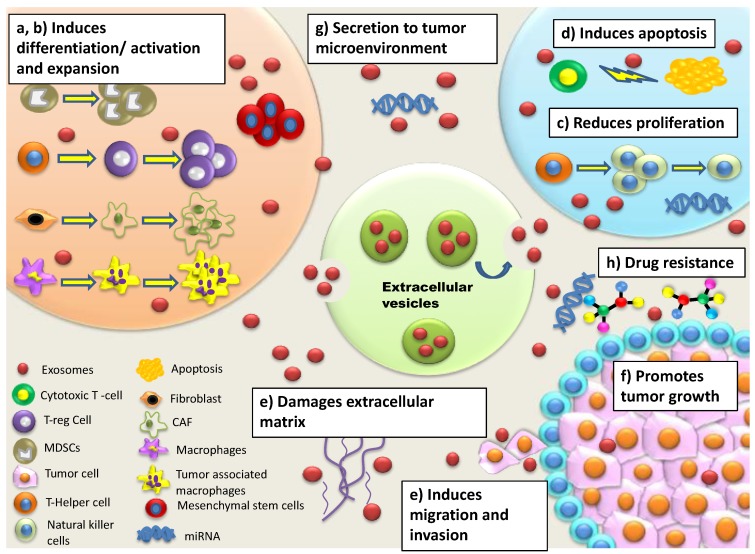
Re-modeling of tumor microenvironment by EVs in CRC. EVs are associated with inducing re-modelling of the tumor microenvironment to promote tumor progression via mechanisms including; (**a**) differentiation of macrophages, stromal fibroblasts etc., (**b**) secretion of cytokines/chemokines to induce expansion of regulatory cells such as Tregs, myeloid derived secreted cells (MDSCs), (**c**) reducing proliferation of natural killer cells (NK cells), (**d**) inducing apoptosis of cytotoxic T-cell (CD8+), (**e**) damaging extracellular matrix to promote cell migration and invasion, (**f**) inducing phenotypic changes, (**g**) secreting non-coding RNAs, (**h**) promoting drug resistance.

**Figure 2 cancers-11-00746-f002:**
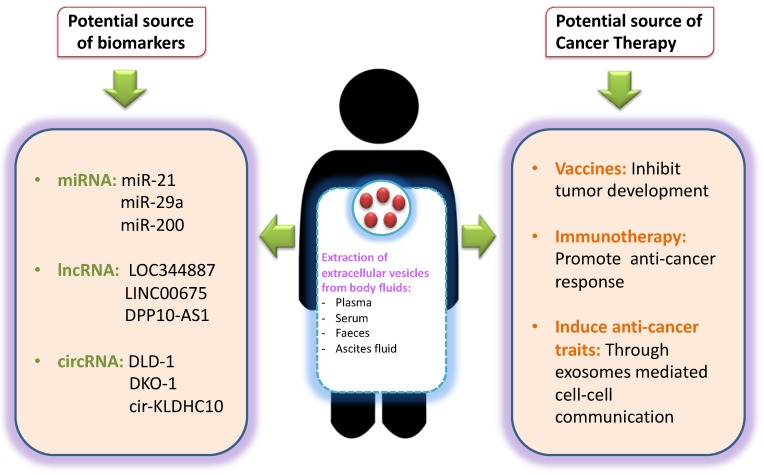
EVs as a potential source of biomarkers and therapeutic targets in CRC: EVs are secreted in various body fluids including plasma, serum, faeces ascites fluid etc. [150]. The detection of cancer associated EVs and its cargoes including miRNA, lncRNA, circ. RNA and several proteins can be used as potential source of biomarkers/prognostic markers in CRC. In cancer therapeutics, engineered EVs can be used for potential vaccine development and induction of immuno-modulatory mechanisms to promote anti-tumor response such as depletion of Tregs and MDSCs, increasing proliferation of NK cells, monitoring of cytokine/chemokines and utilizing EV cargoes as nanoparticles for drug delivery.

**Table 1 cancers-11-00746-t001:** Mechanisms of tumor microenvironment modulation by Colorectal cancer-associated extracellular vesicles.

Mechanisms	Potential Role in Tumour Progression	Ref.
A. Immune system regulation and modulation		
Reduced activation, cytotoxicity, proliferation of Natural Killer (NK) cells	Increased secretion of immunosuppressive factors leads to resistance in NK cells against inducing Fas- or perforin-mediated apoptosis Secretion of TGF-β1 disrupts IL-2 signaling to NK cells Diminished expression of NKG2D receptor on NK cells reduces recognition of malignant cells	[37,38,39]
Induction of CD4+ and CD8+apoptosis	Activation of apoptosis-inducing ligands (Fas ligand, TNF-related apoptosis-inducing ligand (TRAIL), galectin 9 etc. induces apoptosis of T cells Inhibition of IL-2-dependent CD8+ T cell activation leads to immunosuppression	[29,39,40]
Modulation of Tumour associated macrophages (TAMs), Cancer associated fibroblast (CAFs), epithelial and endothelial target cells	Enhances release of cytokines/chemokines, immunosuppressive factors, immune checkpoint modulators, Tregs, myeloid-derived suppressor cells (MDSCs) and type 2 helper (Th2) T cells that directly suppress activated T cells	[41]
Increased secretion of myeloid derived suppressor cell (MDSC) and T regulatory cells (Treg)	Elevated levels of MDSCs and Tregs lead to secretion of cytokines/chemokines and activation of signalling pathways such as MAPK and NF-κB leading to immunosuppression T regs suppress activation, proliferation and effector functions of CD4+, CD8+ cells, NK cells, B cells and dendritic cells (DC) Increased Treg levels initiate angiogenesis and increased vascularization	[39,42,43]
Alteration of complement system	Increased expression of surface molecules CD55, CD59, CD9 signals leads to inhibition of complement-mediated lysis	[44]
Modulation of mesenchymal stem cells	Leads to atypical morphology/spheroid formation that induces acidic microenvironment and leads to immuno-suppression Spheroids activate NOTCH3/Wnt signalling pathways that facilitates drug resistance	[40,45]
Enhanced release of reactive oxygen species and Nitric oxides	Leads to induction of hypoxic environment facilitating immuno-suppression	[46]
Interaction of miRNA with mRNA transcripts	Leads to mRNA deregulation converting them into oncogenes or tumor suppressors Increases the proliferation of fibroblasts, epithelial and endothelial target cells leading to immune suppression and induction of drug resistance	[47,48]
B. Metastasis		
Expression of adhesion molecules/integrins	Facilitates metastatic organotropism/establishment of pre-metastatic niche via adhesion with specific resident cells/fibroblasts/epithelial cells for invasion	[49,50]
Upregulation of cytoskeleton re-modelling Secretion of extracellular matrix associated proteins Induction of epithelial-mesenchymal transition (EMT)	Leads to formation of invadopodia for increased motility Causes vascular leakiness Releases EVs-bound proteins for enhanced migration of tumor cells	[51,52]
Transfer of miRNAs to immune cells	Leads to macrophages skewing towards pro-inflammatory phenotype facilitating colonization and invasiveness	[30]
Modulation of tumour microenvironment	Leads to macrophage infiltration, activation of Src phosphorylation signaling pathways, release of pro-inflammatory cytokines and chemokines for extracellular matrix re-modelling CAFs induce motile and proteolytic activity in tumor cells for enhanced invasiveness	[53]
Modulation of mesenchymal stem cells	Leads to atypical morphology/spheroids formation inducing immuno-suppression and proliferation/invasion	[30]
C. Drug Resistance		
Sequestering, expulsion of cytotoxic drugs	EVs extrude cytotoxic drugs EVs compete with anti-tumor drugs for their binding sites leading to drug resistance	[54,55]
Induction of Stem-ness in tumor cells	Induces atypical morphology and spheroid formation leading to immuno-suppression and expression of drug resistant phenotypes Modulation of CAFs by cancer stem cells leads to increased Wnt activity in cancer stem cells conferring drug resistance	[56,57]
Activation of signalling pathways	STAT1 dependent response and NOTCH3 signalling pathway leads to decreased cell apoptosis and chemo-resistance	[57,58]
Lateral transfer of drug resistant phenotypes via miRNAs	miRNAs carrying drug resistant phenotypes confer chemo resistance by transfer of resistant genes to sensitive recipient tumor cells	[16]

NK: Natural Killer cells; TGF-β1: Transforming growth factor beta 1; IL-2: Interleukin-2; NKG2D: Natural Killer Group 2D; TRAIL: TNF-related apoptosis-inducing ligand; Tregs: T regulatory cells; MDSC: myeloid-derived suppressor cells; Th2: type 2 helper cells; MAPK: Mitogen-activated protein kinases; NF-κB: Nuclear factor kappa-light-chain-enhancer of activated B cells; DC: dendritic cells; CAFs: Cancer associated fibroblasts.

**Table 2 cancers-11-00746-t002:** Key targets and role of EV derived non-coding RNAs in colorectal cancer.

micro-RNA/ lncRNA/Proteins	Level	Key Targets	Role in CRC	Reference
miR-17-92a	upregulated	TSP-1, CTGF, PTEN,BCL2L11,E2F1,E2F2, E2F3,TGFBR2,CDKN1A,BIM	Angiogenesis, proliferation, metastasis	[71,72,73]
miR-18a	upregulated	ATM, mTORC1, hnRNP A1,CDC42	Cell proliferation, migration	[71,73,74,75]
miR-92a	upregulated	PTEN	cell proliferation, migration, invasion, prognosis	[76]
miR-218	downregulated	BMI-1	cell proliferation, apoptosis, cell cycle arrest	[77]
miR-31	upregulated	CDKN2B, RASA1,FIH-1, RhoBTB1	Cell proliferation, invasion, migration, tumor growth, prognosis	[73,78,79]
miR-95	upregulated	SNX1	cell proliferation, tumor growth	[80]
miR-29a	upregulated	KLF4	cell invasion, metastasis, prognosis	[81,82]
miR-96	upregulated	TP53INP1, FOXO1, FOX03a	cell proliferation	[83]
miR-214	downregulated	FGFR1	cell proliferation, migration, invasion	[84]
miR-100	downregulated	RAP1B	cell proliferation, invasion, apoptosis	[79]
miR-194	downregulated	PDK1, AKT2, XIAP, MAP4K4	Cell proliferation, apoptosis, migration, prognosis	[79]
miR-206	downregulated	NOTCH3	cell proliferation, migration, apoptosis, cell cycle arrest	[85]
miR-103	upregulated	DICER, PTEN	cell proliferation, migration, tumor growth	[86]
miR-143	upregulated	KRAS, ERK5, MACC1, HK2, IGF1R, DNMT3A	Cell proliferation, metastasis	[73,74]
miR-196b	upregulated	FAS	cell apoptosis	[87]
miR-148a	upregulated	BCL2	Cell proliferation	[73,88]
miR-375	downregulated	PIK3CA	cell proliferation, cell cycle arrest, tumor growth	[89]
miR-181a	upregulated	WIF-1, PTEN	cell proliferation, migration, invasion, prognosis, tumor growth	[90,91]
miR-21	upregulated	PDCD4, CCL20, Cdc25A, TGFBR2, PTEN, RHOB, RASA1	Cell proliferation, migration, invasion, metastasis, stemness	[73,92]
miR-155	upregulated	CLDN1	Cell proliferation, migration, invasion, chemoresistance	[73]
miR-145	downregulated	FASCIN-1	Cell proliferation, invasion, tumor growth	[93]
miR-32	upregulated	PTEN	cell proliferation, migration, invasion, apoptosis	[94]
miR-378	downregulated	VIMENTIN	cell proliferation, invasion, tumor growth, prognosis	[95]
miR-124	downregulated	STAT3	cell proliferation, apoptosis, tumor growth, prognosis	[96]
miR-126	downregulated	VEGF, IRS-1, CXCR4	cell proliferation, migration, invasion, prognosis	[97,98,99,100]
miR-10b	upregulated	Syndecan-1	tumor suppression, invasion	[101]
miR-375	upregulated	Bcl-2	apoptosis, tumor suppression	[102]
miRNA-210	upregulated	VMP1	tumor growth, metastasis, migration, invasion	[103,104]
miR-200c	upregulated	ZEB2 and SNAI, PTEN	migration, metastasis, invasion, tumor growth	[102,105]
miR-141	upregulated	ZEB2 and SNAI, MAP2K4	migration, metastasis, cell proliferation	[102,106]
miR-429	upregulated	ZEB2 and SNAI, HOXA5	migration, metastasis, tumor growth	[102,107]
miRNA-19a	upregulated	TIA1	Cell proliferation and migration	[108,109]
let-7a	upregulated	NIRF	Cell proliferation, tumorigenesis	[108,110]
miR-1246	upregulated	CycG2, CCNG2	differentiation, invasion, metastasis, chemoresistance	[108,111]
miR-150	upregulated	iASPP	tumor growth, metastasis, cell proliferation	[108,112]
miR-21	upregulated	TIMP3, ANP32A, THRB, PELI1, SPRY 1/2, PDCD4, FASL, PTEN, BCL2, PPARA, HNRPK, TP63	Cell proliferation, invasion, metastasis, apoptosis	[108,113]
miR-223	upregulated	STMN-1	cell proliferation, migration, invasion	[108,114,115]
miR-23a	upregulated	MTSS1	migration, metastasis, cell migration, invasion	[108,116]
miR-193a	upregulated	Caprin1	tumor progression	[117]
ΔNp73	upregulated		Cell proliferation, drug resistance	[118]
CRNDE-h	upregulated	DUSP5/CDKN1A	proliferation, apoptosis, metastasis	[119,120]
MAGEA3	upregulated		cell growth, differentiation, invasion	[121,122]

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
