# Peer review of "The Role of Extracellular Vesicles as Modulators of the Tumor Microenvironment, Metastasis and Drug Resistance in Colorectal Cancer"

_cancers, 2019, doi:10.3390/cancers11060746_

Reviewer 1 Report

This article might become a first comprehensive review of Extracellular vesicles focusing on colorectal cancer.  However, I feel that the description of EV involvement are too brief and largely missing from the manuscript, especially in chapter 5 and 6.

To improve the manuscript, I will suggest following:

·       Describe differences between Exosomes and Microvesicles as in molecular composition including distinct protein markers and lipid composition and comment on why and how they are collectively termed EVs.

·       You may want to include a brief description of subpopulations of EVs, recently published by Jan Lötvall (PMID: 29432782) and comment on methods for EV isolation and technical hurdles. (could be in conclusion if not in this part)

·       The technical challenges of EV use as a biomarker should be discussed in detail in “7. challenge of the use EVs as potential source of biomarkers in CRC”.

Author Response

This article might become a first comprehensive review of Extracellular vesicles focusing on colorectal cancer.  However, I feel that the description of EV involvement are too brief and largely missing from the manuscript, especially in chapter 5 and 6

Author Response (AR):  As per the reviewer’s suggestion we have extended the information on the involvement of EVs in section 5 and 6 in the revised manuscript.

To improve the manuscript, I will suggest the following:

Describe differences between Exosomes and Microvesicles as in molecular composition including distinct protein markers and lipid composition and comment on why and how they are collectively termed EVs

AR: As reviewers suggested, we have added the description on the molecular composition of the exosomes and microvesicles in chapter 2 of the revised manuscript as well as commented on the reason why they are collectively called EVs

You may want to include a brief description of subpopulations of EVs, recently published by Jan Lötvall (PMID: 29432782) and comment on methods for EV isolation and technical hurdles. (could be in conclusion if not in this part)

AR: As the reviewer suggested we have incorporated subpopulation of EVs in the revised manuscript.

The technical challenges of EV use as a biomarker should be discussed in detail in “7. challenge of the use of EVs as a potential source of biomarkers in CRC”.

AR:  We have added a section (8.0)  of the challenge of the use of EVs as a potential source of biomarkers in CRC”.

Reviewer 2 Report

The authors presented an extensive review on the role of extracellular vesicles for tumor microenvironment, metastasis and drug resistance in colorectal cancer.

The ms is well organized in detaled manner.

Minor corrections are needed.

 As for table 1., contents of classified columns and cells are not well-defined. Modification of contents or addition of reference number will be helpful.  

In line 354 of page10, (ref) need correction.

In line 385 of page 11, DLD-1 and DKO-1 are name of cell line. Correction or removal are needed.

Author Response

The authors presented an extensive review on the role of extracellular vesicles for tumor microenvironment, metastasis and drug resistance in colorectal cancer.

AR: We appreciate reviewer’s positive comments regarding our manuscript and mentioning that “The authors presented an extensive review on the role of extracellular vesicles----“

The ms is well organized in detaled manner

AR: We are very thankful to the reviewer for positive feedback

Minor corrections are needed

AR: Bellow we are providing point by point the minor corrections suggested by the reviewer.

As for table 1., contents of classified columns and cells are not well-defined. Modification of contents or addition of reference number will be helpful

AR: We are very thankful to reviewer for this point. We have revised Table 1 with the new column with reference citation.

In line 354 of page10, (ref) need correction

AR: Reference has been cited in the revised manuscript.

In line 385 of page 11, DLD-1 and DKO-1 are name of cell line. Correction or removal are needed

AR: We have removed DLD-1 and DKO-1 from the manuscript as correctly pointed by the reviewer.

Round  2

Reviewer 1 Report

The manuscript improved however there are some statements to be re-edited.

115: Authors state “Exosomes are usually rich in tetraspanins, integrins, cytokines and rich in ceramides 115 and cholesterol…” This statement applies to other types of EVs. If they are specific to Exosomes but not to other types of EVs, please clarify and add reference(s).

116: Authors claim “less amount of phosphatidyl serine”. Exosomes are smaller EVs so that they contain less amount of phosphatidylserine (PS), which could be just a common sense? If there are any studies showing that exosomes contain lower ratio of PS per vesicles compared to other types, please clarify and add reference(s).

Author Response

15: Authors state “Exosomes are usually rich in tetraspanins, integrins, cytokines and rich in ceramides 115 and cholesterol…” This statement applies to other types of EVs. If they are specific to Exosomes but not to other types of EVs, please clarify and add reference(s).

116: Authors claim “less amount of phosphatidyl serine”. Exosomes are smaller EVs so that they contain less amount of phosphatidylserine (PS), which could be just a common sense? If there are any studies showing that exosomes contain lower ratio of PS per vesicles compared to other types, please clarify and add reference(s)

Authors Response: We are very thankful to reviewer for pointing out this misleading sentence. We have modified the sentence as following. We have also provided a reference.  

“All EVs are usually rich in tetraspanins, integrin’s, transferrin receptors, LAMP1/2, heparan sulfate proteoglycans, EMMPRIN (BSG); ADAM10; GPI-anchored 5ʹnucleotidase CD73 (NT5E), complement-binding proteins CD55 and CD59, sonic hedgehog (SHH), ESCRT-I/II/III, heat shock proteins, immune cells/ MHC class I, cytokines, ceramides and cholesterol[18]